



# Undetected BVOCs from Norway spruce drive total ozone reactivity measurements

Steven Job Thomas[1,2], Toni Tykkä[1], Heidi Hellén[1], Federico Bianchi[2], and Arnaud P. Praplan[1]

[1]Atmospheric Composition Research, Finnish Meteorological Institute, Helsinki, 00101, Finland.
[2]Institute for Atmospheric and Earth System Research/Physics, Faculty of Science, University of Helsinki, Helsinki, 00560, Finland.

**Correspondence:** Steven Job Thomas (steven.thomas@helsinki.fi)

**Abstract.** Biogenic Volatile Organic Compounds (BVOCs) are continuously emitted from terrestrial vegetation into the atmosphere and react with various atmospheric oxidants, with ozone being an important one. The reaction between BVOCs and ozone can lead to low volatile organic compounds, other pollutants, and the formation of secondary organic aerosols. To understand the chemical and physical processes taking place in the atmosphere, a complete picture of the BVOCs emitted is
necessary. However, the large pool of BVOCs present makes it difficult to detect every compound. The total ozone reactivity method can help understand the ozone reactive potential of all BVOCs emitted into the atmosphere and also help determine if current analytical techniques can measure the total BVOC budget.

In this study, we measured the total ozone reactivity from the emissions of a Norway spruce tree in Hyytiälä in late summer using the Total Ozone Reactivity Monitor (TORM) built at the Finnish Meteorological Institute (FMI). We also conducted
chemical characterisation and quantification of the BVOC emissions using a gas chromatograph coupled with a mass spectrometer (GC-MS).

The measured total ozone reactivity reached up to $7.3 \times 10^{-9}\ \mathrm{m^3\,s^{-2}\,g^{-1}}$, which corresponds to $64\ \mathrm{\mu g\,g^{-1}\,h^{-1}}$ of $\alpha$-pinene. Stress-related sesquiterpenes such as $\beta$-farnesene and $\alpha$-farnesene, and an unidentified sesquiterpene contributed the most to the observed emissions. However, the observed emissions made up only 35% of the measured total ozone reactivity, with
sesquiterpenes being the most important sink for the ozone. High total ozone reactivity was especially seen during high temperature periods, with up to 95% of the reactivity remaining unexplained. Emissions of unidentified stress-related compounds could be the reason for the high fraction of missing reactivity

## 1   Introduction

Terrestrial vegetation is the largest emitter of biogenic volatile organic compounds (BVOCs) which are produced from a variety of sources but with most emissions observed from foliage. BVOCs play an important role in the global climate and influence air quality by acting as precursors to ozone and secondary organic aerosols (SOA) (Šimpraga et al., 2019). Thousands of



compounds are emitted into the atmosphere, but a group of compounds called terpenoids namely isoprene and monoterpenes (MTs) are the dominant global BVOCs. According to the most up-to-date biogenic emission model, Model of Emissions of

Gases and Aerosols from Nature (MEGAN v2.1; Guenther et al., 2012), BVOCs from tropical areas are responsible for 80% of the terpenoid emissions and 50% of other emissions while trees from other biomes contribute to 10% of the total BVOC emissions. Another study (Messina et al., 2016) estimates that terpenoid emissions especially MTs and sesquiterpenes (SQTs) from boreal forests in the northern regions are higher than calculated using MEGAN model.

The boreal zone is one of the most active zones for new particle formation and the northern boreal region has been extensively

studied for aerosol formation (e.g., Tunved et al., 2006; Kulmala et al., 2013; Ehn et al., 2014; Kerminen et al., 2018; Barreira et al., 2021). There is substantial evidence to prove that terpenoids are one of the main precursors for aerosol formation in several sites (Tunved et al., 2006; Barreira et al., 2021). Terpenoids such as monoterpenes and sesquiterpenes are highly reactive and once emitted into the atmosphere will undergo oxidation via reactions with oxidants such as the hydroxyl radical (OH), the nitrate radical ($NO_3$), or ozone ($O_3$) to form low volatile oxidised products that participate in atmospheric particle

formation (Paasonen et al., 2013; Ehn et al., 2014; Jokinen et al., 2015; Bianchi et al., 2019). Several studies conducted in the boreal forest in Finland (Hyytiälä) have observed monoterpenes like $\alpha$-pinene on ozonolysis or OH initiated reactions to form highly oxygenated molecules that may lead to the formation of SOA (Ehn et al., 2012; Bianchi et al., 2017). Since $\alpha$-pinene is one of the most prominent BVOCs to be emitted globally it is also used as a proxy to predict various atmospheric processes (Guenther et al., 2012; Holopainen et al., 2017).

However, BVOC emissions from plants can be complex, especially when exposed to stress like heat and drought. The emission blend and quantity can vary depending on the plant species, organ of the plant, location and other environmental factors. With the help of recent advances in BVOC measurement techniques, high emissions of sesquiterpenes and low volatile oxygenated compound emissions have been detected from trees (Hellén et al., 2021; Hakola et al., 2023). Some of the emitted sesquiterpenes like $\beta$-caryophyllene have been reported to have prominent effects such as higher SOA mass yields, and a larger

impact on ozone chemistry (Faiola et al., 2018; Hellén et al., 2018; Ylisirniö et al., 2020; Barreira et al., 2021). Consequently, the volatile bouquet of plant emissions contains a complex mixture of organics—many of which can act as precursors to SOA.

Despite more than 1700 BVOCs being identified over the past few decades, total OH reactivity studies conducted across various sites have shown that there still lies a major fraction of OH reactivity (Dudareva et al., 2013; Yang et al., 2016) that cannot be identified with known BVOCs. Praplan et al. (2020) found unexplained OH reactivity to be as high as 96%

from emissions of birch tree, and up to 82% for pine and spruce emissions. The unexplained fraction was high especially when stress-induced compounds such as green leaf volatiles (GLVs) were emitted more from the trees. Nölscher et al. (2013) showed that 15% to 84% of total OH reactivity from Norway spruce emissions could not be explained. However, another OH reactivity study observed that the total OH reactivity could be explained from the detected isoprene, monoterpene and sesquiterpene emissions (Kim et al., 2011). OH-initiated oxidation of BVOCs could contribute to SOA formation; therefore,

characterising the missing fraction could modify the generalised properties of the ambient air and lead to better understanding of the atmosphere. While the OH radical is a very reactive molecule, it also reacts with most identified BVOCs quite instantly, making the study quite demanding.



Ozone is another important oxidant that is present in the atmosphere which is prominent even in the night unlike OH. But ozone is selectively reactive, i.e. it reacts with molecules containing C-C double bond. Total ozone reactivity studies can help narrow down the reactive compounds that might be emitted from plants. Comparing the directly measured total ozone reactivity (or total ozone loss rate) in the BVOC sample (emissions or ambient) with reactivity derived from known chemical composition of the same BVOC sample will help identify the knowledge gaps in the BVOC compositions. Moreover, ozone reaction with BVOCs has been identified as a source of aerosols (Kulmala et al., 2004; Kammer et al., 2018; Rose et al., 2018). Similar to total OH reactivity, total ozone reactivity can be an important parameter to identify the contribution of BVOCs to atmospheric chemistry and a tool to assess the exhaustiveness of BVOC measurements.

In this study, the Total Ozone Reactivity Monitor (TORM) built at the Finnish Meteorological Institute based on the work by Helmig et al. (2022) was deployed in the field. Simultaneous BVOC and total ozone reactivity measurements were conducted from Norway spruce emissions from August to September 2021. Norway spruce is one of Finland's most common tree species, and their emission characteristics have been studied plenty (Bourtsoukidis et al., 2013; Hakola et al., 2017, 2023). Hakola et al. (2017) has also described that SQT emissions from spruce greatly impact ozone chemistry. Using TORM coupled with emission measurements, the effect of BVOCs from Norway spruce on ozone chemistry can be verified.

## 2 Methodology

The study was conducted at the SMEAR II (Station for Measuring Forest Ecosystem-Atmosphere Relationships II) research station, a boreal forest site, in Hyytiälä, southern Finland (61°51´ N, 24°17´ E, 181 m.a.s.l; Hari and Kulmala (2005)). It is a flagship station maintained by the University of Helsinki where continuous and comprehensive measurements are conducted to study the biosphere-atmosphere interaction. The forest stand is dominated mainly by Scots pine (*Pinus silvestris*), but other species such as Norway spruce (*Picea abies*) and birches are found in the minority.

The instruments to measure BVOCs and total ozone reactivity were placed in a container owned by the Finnish Meteorological Institute, which is located about 128 m south of the SMEAR II mast. The Norway spruce tree used in this study is ca. 50 years old and located 5 m from the container.

### 2.1 Description of branch scale BVOC measurements

The sampling technique and materials used for BVOC emission measurements followed the method described in Hakola et al. (2017). The branch to be measured was placed in a 6 L branch enclosure covered with a transparent fluorinated ethylene propylene (FEP) film. The film was attached to the branch on one end and to a Teflon frame consisting of ports for zero-air inlet, sample-air inlet, and temperature-humidity sensors. Air devoid of any VOC supplied by a zero-air generator (HPZA-7000, Parker Balston, Lancaster, NY, USA) was directed to the enclosure at approximately $4 \, \text{L} \, \text{min}^{-1}$. The relative humidity (RH) and the temperature in the enclosure were recorded with a USB data logger (EL-USB-2 Data Logger, Lascar Electronics, Salisbury, United Kingdom). From 23 August, leaf level sensors were used to measure temperature and RH. A leaf-&-air-temperature conifer type sensor (LAT-C; Ecomatik GmbH, Dachau, Germany) recorded the temperature inside the enclosure





as well as the difference between the needle surface temperature and the air temperature in the enclosure; two HygroVUE5
temperature and relative humidity sensors (Campbell Scientific, Inc., Logan UT, USA) were monitoring these parameters both
inside and outside the enclosure. These sensors output absolute leaf temperature and can help to improve the precision of the
data that is to be measured. A Quantum Sensor SQ-110 SS (Apogee Instruments, Inc., Logan UT, USA) measured the incoming
photosynthetically active radiation (PAR) above the enclosure. After 23 August, the PAR measurements were low throughout

as the new sensor (SQ-110 SS Apogee PAR Quantum Sensor solar calibrated) was placed in a position where it was shaded
by a scaffolding in front. All data reported in this study are given in the time zone of UTC+2, which corresponds to Finnish
winter time.

## 2.2  Total Ozone Reactivity

Total ozone reactivity ($R_{O_3}$) is the inverse of the ozone loss rate in the presence of gases (here, BVOCs) in a sample air. The

total ozone reactivity can be theoretically calculated as the product of the sum of the concentration of individually measured
compounds [$A_i$] and their respective reaction rate coefficient ($k_{O_3+A_i}$) with $O_3$:

$$R_{O_3,\text{calculated}} = \sum_i [A_i] \cdot k_{O_3+A_i} \tag{1}$$

By comparing the theoretically calculated reactivity with measured reactivity, we can assess if all BVOCs from emissions
have been characterised and quantified by the gas chromatograph coupled with mass spectrometer (GC-MS).

### 2.2.1  Total Ozone Reactivity Monitor (TORM)

The total ozone reactivity monitor (TORM) is the instrument devised to experimentally determine total ozone reactivity. The
measurement principle of TORM and the instrument is described in Helmig et al. (2022). The principle of TORM is that the
sample air is enriched with ozone and directed to a reaction chamber where the mixture is allowed to react for a known amount
of time. The difference in ozone before and after the reaction is used to calculate total ozone reactivity using the formula:

$$R_{O_3} \approx \frac{[O_3]_o - [O_3]_t}{[O_3]_o} \frac{1}{\Delta t} = \frac{\Delta[O_3]}{[O_3]_o \Delta t} \tag{2}$$

where $[O_3]_o$ is ozone concentration before reaction and $[O_3]_t$ is ozone concentration after reaction with BVOCs at residence
time $\Delta t$ in the reactor (120 s). The differential signal, $[O_3]$, is directly measured by a modified ozone analyser (see below).

The schematic of TORM used in this study is illustrated in Fig. 1. In brief, TORM consists of three main parts i) reactor ii)
ozone analyser (differential analyser) and generator iii) second ozone analyser (monitor). The sample air enriched with ozone

is directed to the 6 L reaction chamber made of 3 glass flasks (2 L each) connected in series. A 100 ppb of ozone produced by
the ozone generator, which is constantly monitored by the ozone monitor is mixed with sample air containing BVOCs. With
the help of a Python programme, the ozone concentration was maintained at 100 ppb by automatically adjusting the UV lamp
level when the ozone concentration changed beyond ± 2 ppb. Another ozone monitor termed differential analyser, measures



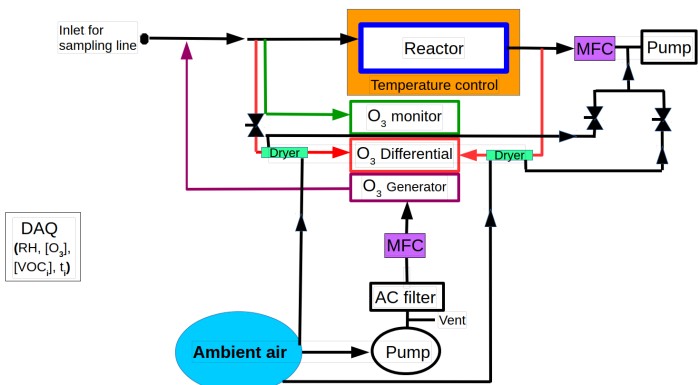

**Figure 1.** Schematic diagram of total ozone reactivity monitor

the difference in ozone concentration before and after the $O_3$-BVOC reaction in the reactor directly. The differential analyser is

a modified configuration of the standard ozone analyser. In the standard configuration, the ozone monitor measures by finding

the difference in ozone concentration between the sample line (ozone-containing air) and the reference line (a line containing

a scrubber to make ozone-free air). In the differential configuration, the scrubber in the reference line is removed to convert it

into a normal sampling line. By connecting one of the lines before the reactor and the other line after, the resulting output from

this differential analyser is the ozone lost in the reactor.

The loss of ozone on the reactor wall was considered by bypassing the branch chamber and connecting the TORM sampling

line to zero air. If the differential signal due to the ozone loss on the reactor wall is $\Delta[O_3]_{\text{zero}}$, then the corrected differential

signal is:

$$\Delta[O_3]_{\text{corr}} = \Delta[O_3] - \Delta[O_3]_{\text{zero}} \tag{3}$$

$$R_{O_3,\text{corr}} = \frac{\Delta[O_3]_{\text{corr}}}{[O_3]_o \Delta t} \tag{4}$$

TORM was calibrated twice using a standard mixture (National Physical Laboratory, U.K.). The mixture contained 200

ppb of $\alpha$-pinene, 3-carene, limonene and 1,8-cineole each. Calibration was performed at the end of the campaign on two

different days. Figure B1 in the appendix shows the calibration lines and that TORM underestimated reactivity at three different

concentrations.

$$R_{\text{torm}} = \frac{R_{O_3,\text{corr}} + 3.15\text{x}10^{-5}}{0.79} \tag{5}$$

Also, due to the mixing of sample air with ozone before reaching the reactor, the total ozone reactivity measured in the

reactor is less compared to total ozone reactivity measured in the sample, which requires correcting $R_{O_3,\text{corr}}$ using the dilution



factor D

$$D = \frac{f_{sample}}{f_{sample} + f_{O_3}} \tag{6}$$


$$R_{sample} = R_{torm}/D \tag{7}$$

where $f_{sample}$ and $f_{O_3}$ are flow rates of sample air and from ozone generator, respectively. Total ozone reactivity of the emissions is normalised using the total air flow through the enclosure (f) and the dry mass of the needles in the enclosure ($m_{dw}$) as in the following formula:

$$R_{O_3,spruce} = R_{sample} \cdot f/m_{dw} \tag{8}$$


### 2.3 In-situ emission measurements

In order to calculate the theoretical reactivity, it is necessary to quantify and identify the chemical composition of BVOCs emitted from the tree. This was made possible with the help of a gas chromatograph coupled with mass spectrometer (GC-MS) which is described previously in Hellén et al. (2018) and Helin et al. (2020). The compounds were collected at $40 \text{ mL min}^{-1}$

for 30 minutes in the cold trap (Carbopack B/Tenax TA) of the thermal desorption unit (TurboMatrix, 350, Perkin-Elmer) connected to the GC (Clarus 680, Perkin-Elmer) coupled with the MS (Clarus SQ 8 T, Perkin-Elmer). A DB-5MS column (60 m, i.d. 0.25 mm, film thickness 0.25 $\mu$m) was used for separation. The instrument was calibrated for 2-methyl-3-butenol (MBO), mono- and sesquiterpenes using liquid standards in methanol solutions. Isoprene was calibrated using a gaseous standard (National Physical Laboratory, 32 VOC mix at the4 ppbv level). $\alpha$-Farnesene was tentatively identified based on the

mass spectra and the retention indices in the NIST library (NIST/EPA/NIH Mass Spectral Library, version 2.0). $\alpha$-Farnesene and the unknown sesquiterpene were quantified based on the response of $\beta$-caryophyllene while bornyl acetate was quantified as nopinone . The emission rate of BVOCs is calculated based on Hakola et al. (2001).

Similarly as for total ozone reactivity, the calculated ozone reactivity based on the chemical composition of emissions detected using the GC-MS can be calculated using eq.1. The calculated reactivity ($R_{O_3,cal}$) is normalised using the flow

through the enclosure (f) and the dry mass of the needle ($m_{dw}$).

### 3 Results and Discussion

### 3.1 Ambient and chamber environment

The weather during the measurement period varied from cold and humid to pleasantly warm conditions. Ambient mean night time temperature was 4 °C cooler than mean day time (14.3 °C) temperature. Ambient temperatures above 20 °C were seen

mostly until 13 August, with the period maximum reaching on 13 August. Chamber temperatures above 25 °C were seen on 6,



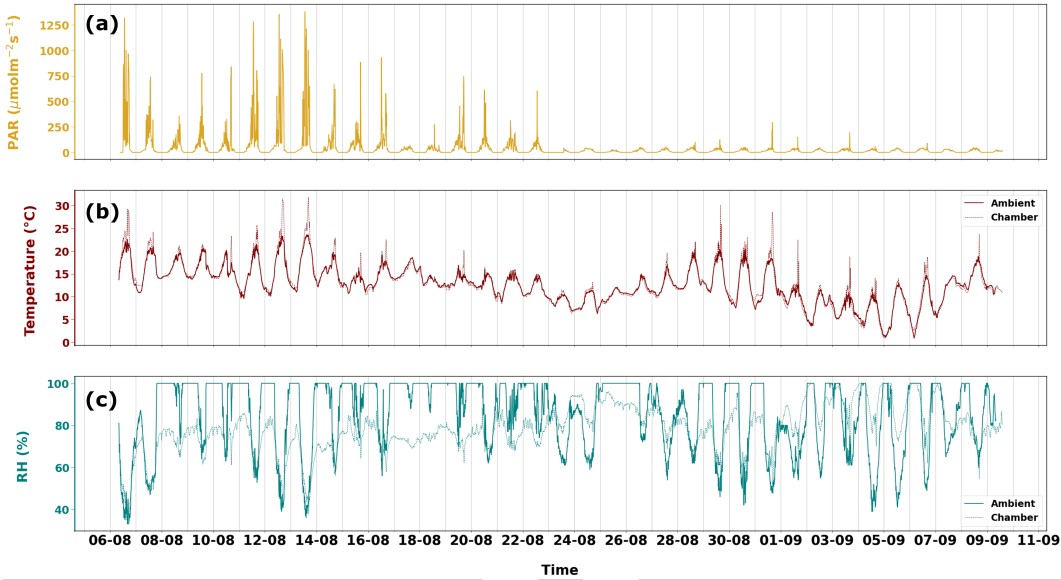

**Figure 2.** The different environmental parameters that were observed from August 6, 2021 to September 10, 2021. In panels (b) and (c), the solid lines are ambient observations downloaded from FMI open weather data. The dashed lines are measurements from the sampling enclosure.

11, 12, 13, 29 and 31 August. High temperatures in the enclosure may have been recorded when prolonged sunlight heated the enclosure, as the effect can be seen before 24 August in Fig. 2. Only 5% of the chamber temperature data deviates from ambient temperature beyond 2 °C. The maximum temperature difference between both is 11.3 °C. These were days when skies were clear or partly cloudy. Mild but frequent precipitation and overcast conditions were also seen during the period. Rainfall was
observed between 8 and 11 August and also every day from 14 to 26 August. While sunny weather and high temperatures were seen from 11 to 13 August and from 29 August to 1 September during which temperature spikes up to 30 °C were measured in the chamber.

**Table 1.** Mean, minimum and maximum of ambient air and branch chamber measurements observed during the campaign

| Parameters | Chamber [a] | | | Ambient | | |
|---|---|---|---|---|---|---|
| | Min | Avg | Max | Min | Avg | Max |
| Temperature (°C) | 0.8 | 12.8 | 31.8 | 0.9 | 12.6 | 23.6 |
| PAR [b] ($\mu$mol m$^{-2}$ s$^{-1}$) | 0 | 39.4 | 1381.6 | | | |
| RH (%) | 36.3 | 78.2 | 100 | 33 | 85.7 | 100 |
| Precipitation (mm) | | | | 0 | 0.16 | 5.9 |

[a] Leaf scale measurements were conducted from August 23, 2021.

[b] PAR was measured by placing the sensor outside on top of the branch enclosure.



## 3.2 Overview of Norway spruce emissions

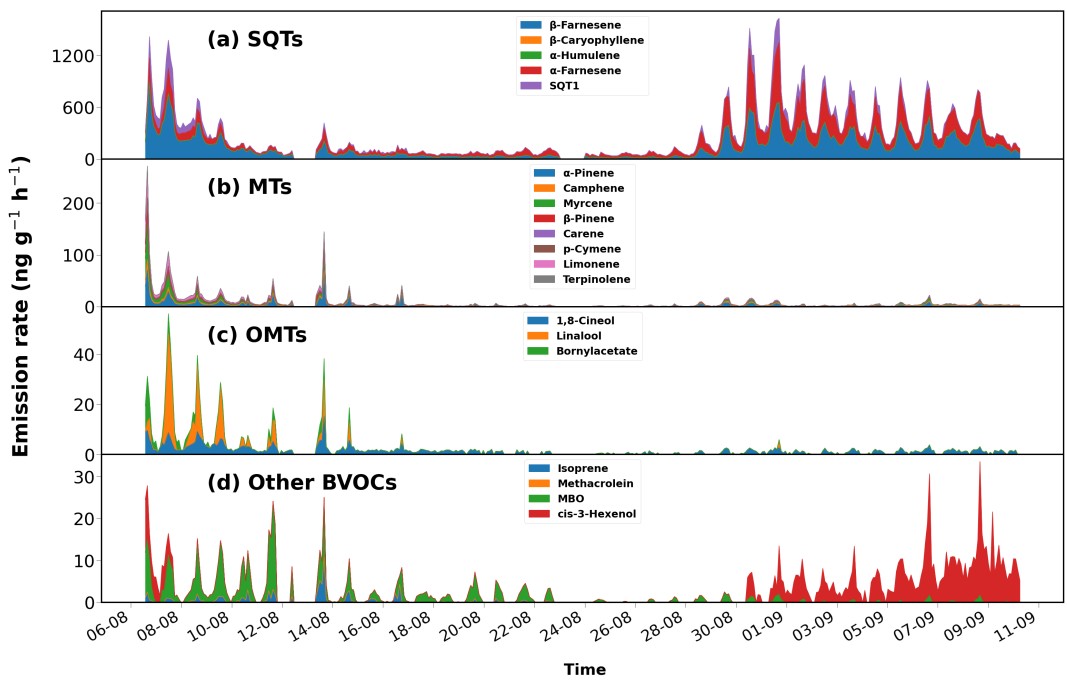

**Figure 3.** Time series of emissions of (a) sesquiterpenes (SQTs), (b) monoterpenes (MTs), (c) oxygenated monoterpenes (OMTs), and (d) different BVOCs observed from Norway spruce.

Different compounds observed during the measurement period and their respective mean emission rates are shown in Appendix A Table A1. Highest emissions were observed from SQTs (Fig. 2) with 281 $\mathrm{ng\,g^{-1}\,h^{-1}}$ as the period average, followed by MTs. The MT emissions observed in our study are lesser than that observed in the study by Hakola et al. (2017) during late summer. In our study, $\beta$-farnesene contributed the highest (137.5 $\mathrm{ng\,g^{-1}\,h^{-1}}$) while comparable emissions of $\alpha$-farnesene (112.4 $\mathrm{ng\,g^{-1}\,h^{-1}}$) were also measured. Minor contributions from certain ozone-reactive compounds like $\beta$-caryophyllene, $\alpha$-humulene, and terpinolene were observed during emission spikes on 7 August and 13 August. The emission pattern observed in this study with high SQTs and lower MTs is in agreement with that observed in Hakola et al. (2017). In that study, the mean emission of $\beta$-farnesene is around 3 times lower than observed here.

A total of 20 compounds were detected, out of which only 7 compounds (4 MTs, 2 SQTs and MBO) were detected to be emitted from the tree everyday. At the start of the measurement period, MT emissions were at their highest, close to 250 $\mathrm{ng\,g^{-1}\,h^{-1}}$, with $\alpha$-pinene and myrcene being the biggest contributors. However, MT emissions decreased gradually with an increase seen only on 13 August, when the temperature reached the maximum. Emissions of MTs were low after 16 August (maximum: 7 $\mathrm{ng\,g^{-1}\,h^{-1}}$) but with clear diel patterns. This similar trend was observed from oxygenated monoterpenes (OMTs) too. Different behaviour was observed from SQTs as their emissions were high during the start and end of the campaign. The magnitude



of the one unidentified sesquiterpene (SQT1) was similar to $\alpha$-farnesene at the beginning. But after 9 August emissions of SQT1 was hardly observed while there were clear emissions of $\alpha$-farnesene. The emissions of $\alpha$-farnesene increased by 2-3 times from 27 August and remained relatively high until the end. We began to observe emissions of SQT1 again starting from 27 August which peaked on 31 August after which it gradually decreased unlike $\alpha$-farnesene. The emissions of $\alpha$-farnesene correlated positively with $\beta$-farnesene (r=0.85) and *cis*-3-hexenol (r=0.57). The three highly emitted SQTs correlated weakly with temperature and PAR (r<0.3) and *cis*-3-hexenol correlated with chamber parameters poorly (r<0). The low correlation also suggests that these compounds may have been emitted as a form of stress. Emissions of $\alpha$-farnesene and $\beta$-farnesene from norway spruce have been reported to be induced as a response to insect infestation (Blande et al., 2009; Kännaste et al., 2009; Kleist et al., 2012). High emissions of these two compounds are a typical reaction of spruce trees to biotic stress. Although potential signals of biotic stress has been noted, there is no evidence to confirm signs of insect infestation. Emissions of the SQT1 could also be a result of some stress that we could not discern in this study.

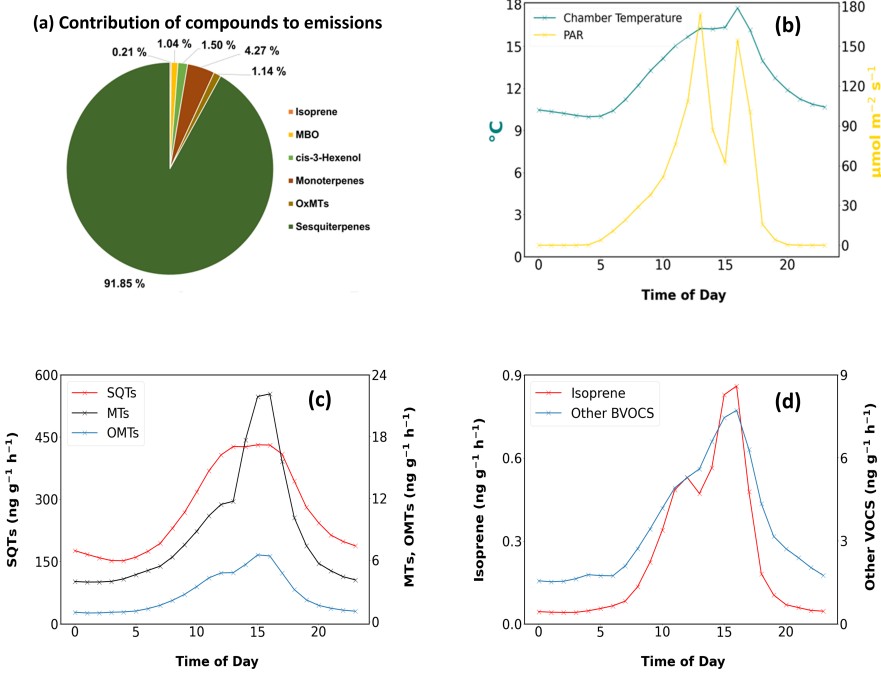

**Figure 4.** (a) Contribution of observed BVOC to Norway spruce emissions. Mean diel variation in emission rates of (b)chamber temperature and PAR, (c) MTs, OMTs, and SQTs and (d) isoprene and the sum of the rest of BVOCs

All compounds besides methacrolein and $\alpha$-humulene showed a clear diel pattern forming a peak post noon. As both those compounds were detected only a couple of times during the whole period, no diel pattern exists for them. The diel variation of emissions showed all groups of compounds besides SQTs to gradually increase over the day and shoot up after 13:00 (Fig. 4). This increase coincides with the high PAR observed at 13:00. SQT emissions that were dominated by $\beta$-farnesene and $\alpha$-farnesene gradually started to increase from 06:00. The SQT emissions remained high and steady from 12:00 to 16:00.



## 3.3 Total ozone reactivity

**Table 2.** $O_3$ reaction rate coefficients used in reactivity calculations. Reaction rate coefficients are taken from IUPAC Task Group on Atmospheric Chemical Kinetic Data Evaluation, (http://iupac.pole-ether.fr, accessed on 27 January 2022), Atkinson et al. (2004, 1990); Calvert et al. (2015); Matsumoto (2016)

| Compound | $k_{O_3}$ (cm$^3$ s$^{-1}$) | T = 298 K |
|---|---|---|
| Methacrolein | $1.40 \times 10^{-15} \cdot e^{(-2100/T)}$ | $1.22 \times 10^{-18}$ |
| MBO | | $1.0 \times 10^{-17}$ |
| cis-3-Hexenol | | $6.40 \times 10^{-17}$ |
| Isoprene | $1.05 \times 10^{-14} \cdot e^{(-1998/T)}$ | $1.28 \times 10^{-17}$ |
| **Monoterpenes** | | |
| $\alpha$-Pinene | $8.22 \times 10^{-16} \cdot e^{(-640/T)}$ | $9.6 \times 10^{-17}$ |
| Camphene | $9.0 \times 10^{-18} \cdot e^{(-860/T)}$ | $5.02 \times 10^{-19}$ |
| Myrcene | $2.69 \times 10^{-15} \cdot e^{(-520/T)}$ | $4.7 \times 10^{-16}$ |
| $\beta$-Pinene | $1.39 \times 10^{-15} \cdot e^{(-1280/T)}$ | $1.9 \times 10^{-17}$ |
| Carene | | $4.9 \times 10^{-17}$ |
| $p$-Cymene | | $5 \times 10^{-20}$ |
| Limonene | $2.91 \times 10^{-15} \cdot e^{(-770/T)}$ | $2.2 \times 10^{-16}$ |
| Terpinolene | | $1.6 \times 10^{-15}$ |
| **Oxygenated monoterpenes** | | |
| 1,8-Cineol | | $1.5 \times 10^{-19}$ |
| Linalool | $1.6 \times 10^{-15} \cdot e^{(-396/T)}$ | $4.2 \times 10^{-16}$ |
| Bornylacetate | | $^a\, 7.0 \times 10^{-20}$ |
| **Sesquiterpenes** | | |
| $\beta$-Farnesene | $1.5 \times 10^{-12} \cdot e^{(-2350/T)}$ | $5.64 \times 10^{-16}$ |
| $\beta$-Caryophyllene | | $1.2 \times 10^{-14}$ |
| $\alpha$-Humulene | | $1.2 \times 10^{-14}$ |
| $\alpha$-Farnesene | $3.5 \times 10^{-12} \cdot e^{(-2590/T)}$ | $5.89 \times 10^{-16}$ |
| Unknown Sesquiterpene (SQT1) | | $^b\, 2.2 \times 10^{-16}$ / $^c\, 1 \times 10^{-14}$ |

$^a$ Reaction rate of camphor (similar structure)

$^b$ Reaction rate of $\alpha$-cedrene.

$^c$ Reaction rate equivalent to a fast reacting compound like $\beta$-caryophyllene.

TORM observed high reactivity from the emissions of norway spruce, with a few peaks during the start and another set of peaks towards the end of August. The measured total ozone reactivity was split into 1) high reactivity (07.08 - 08.08; 12.08



- 13.08; 29.08 - 02.09) and 2) low reactivity (09.08 - 11.08; 14.08 - 28.08; 3.09 - 10.09) periods based on the daily average. The highest total ozone reactivity was seen on 31 August with a value of $7.3 \times 10^{-9} \mathrm{~m^3\,s^{-2}\,g^{-1}}$ (Fig. 5a). This maximum value corresponds to $64 \mathrm{~\mu g\,g^{-1}\,h^{-1}}$ (360 ppb in the enclosure) of $\alpha$-pinene or $0.8 \mathrm{~\mu g\,g^{-1}\,h^{-1}}$ (2.9 ppb in the enclosure) of

$\beta$-caryophyllene. These concentration levels are typically not observed in the atmosphere. These compounds once emitted into the atmosphere react almost instantly with atmospheric oxidants and therefore such high concentrations will not be present in the ambient environment. However high emissions of these compounds can be seen in branch enclosure studies.

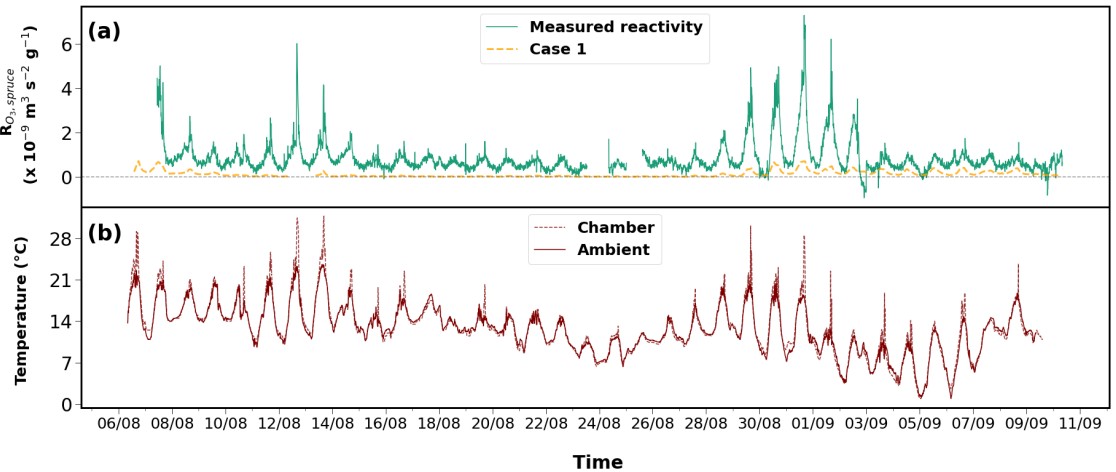

**Figure 5.** Observation of (a) measured total ozone reactivity (green), case 1 (dashed orange line) calculated ozone reactivity using lower reaction rate. (b) Chamber (dashed red) and ambient (solid red) temperatures observed during the measurement period.

To compare the total ozone reactivity with calculated reactivity two different cases have been considered. For each case, the unknown sesquiterpene (SQT1) was assigned a different reaction rate with ozone. In case 1, the reaction rate of cedrene which is on the lower end of the known sesquiterpene reaction rates, was assigned ($2.2 \times 10^{-16} \mathrm{~cm^3\,s^{-1}}$). While in case 2 a faster reaction rate with ozone was used ($1.2 \times 10^{-14} \mathrm{~cm^3\,s^{-1}}$). This reaction rate is equivalent to that of $\beta$-caryophyllene. The two reaction rates will explain how the calculated reactivity compares with the measured total ozone reactivity. After comparing the TORM signal with the two cases, any unexplained signal confirms the presence of unmeasured compounds.

The total ozone reactivity obtained from the differential ozone signal (eq. 4) assumes linear ozone decay (Helmig et al.,

2022). However, the linear relationship becomes invalid as ozone decays exponentially in the presence of a fast reacting compound when concentrations are above the limit of detection of TORM (Fig. B2, Appendix B2). This leads to an underestimation of total ozone reactivity measured by TORM. Therefore in order to compare total ozone reactivity with case 2 of calculated reactivity, a correction factor was applied to the measured total ozone reactivity when concentration of SQT1 increased beyond 0.2 ppb to obtain the true total ozone reactivity in the presence of a fast reacting compound (Fig. 6).

The total ozone reactivity measured by TORM follows the pattern of the calculated reactivity (r=0.50 for case 1 and r=0.52 for case 2) for both cases especially during the later stage of the campaign. From 14 to 28 August, emissions detected from



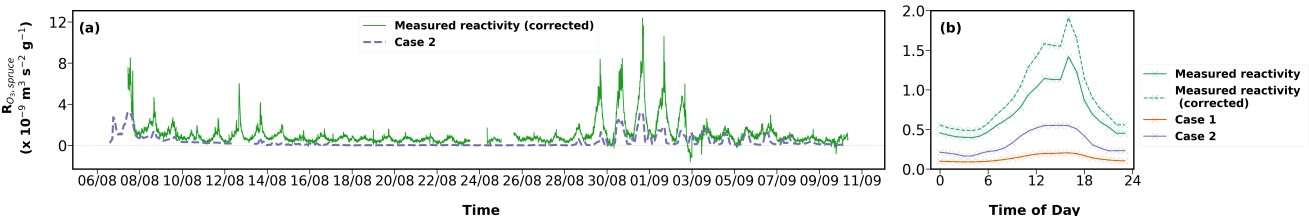

**Figure 6.** Observation of (a) measured total ozone reactivity (green) corrected in the presence of fast reacting compound and case 2 (dashed violet line) calculated ozone reactivity using higher reaction rate. (b) Diel plot of measured reactivity, measured reactivity (corrected), case 1 and case 2.

spruce were close to the detection limit of the GC-MS. The measured reactivity was also close to the detection limit of TORM during this period of 14 days. The period experienced either rainfall or heavily overcast conditions. This long and low reactivity period was immediately followed by high total ozone reactivity for 5 days. Without the low reactivity period between 14 and
28 August, the correlation between measured and calculated total ozone reactivity increased to 0.62 for case 2. Sudden bursts of reactivity like that seen on 31 August followed the temperature spikes observed in the chamber. The sudden increase in temperature may have caused emissions of highly reactive compounds that TORM detected. In Fig. 6, it can be seen that measured and case 2 of calculated reactivity match after 3 September 2021. This would mean that our assumption for SQT1 to be a fast reacting compound is possible and that at least half of the reactivity was driven by SQT1 especially during the last
few days of the campaign.

The diel pattern of measured reactivity increased gradually until 12:00 in the afternoon followed by a sudden peak at 16:00 (Fig. 6b). This coincides with the peak in temperature observed at the same time (Fig. 4b). At 16:00, all observed BVOC classes except SQTs showed a spike in their emission pattern. However, since the calculated reactivities were driven by SQT emissions, there was no unusual spike seen in the calculated reactivities. Instead, in both cases, the reactivity gradually
increased throughout the day and reached the maximum at 16:00. Hence, other reactive compounds were emitted during this hour which were missed out by the GC-MS.

### 3.4 Missing reactivity

The unknown sesquiterpene detected from the emissions cause ambiguity to the calculated reactivity. During the whole campaign, the missing fraction varied from 65%-86% based on the case and period of reactivity. Between the high and low periods,
the missing fraction did not differ by more than 10% indicating the presence of undetected compounds during both the periods.

Calculated reactivity could not explain the observed total ozone reactivity well during the low reactivity period. During this period, the correlation for the two cases with the total ozone reactivity was less than <0.37. Between 14 and 28 August (low reactivity period), the use of either reaction rates did not change the calculated reactivity as the emissions of SQT1 were either very less or absent. However, the low reactivity period in September (3.09 - 10.09) could be explained partly by the calculated
reactivity when the higher reaction rate was applied. At least half of the reactivity was driven by SQTs alone during this low





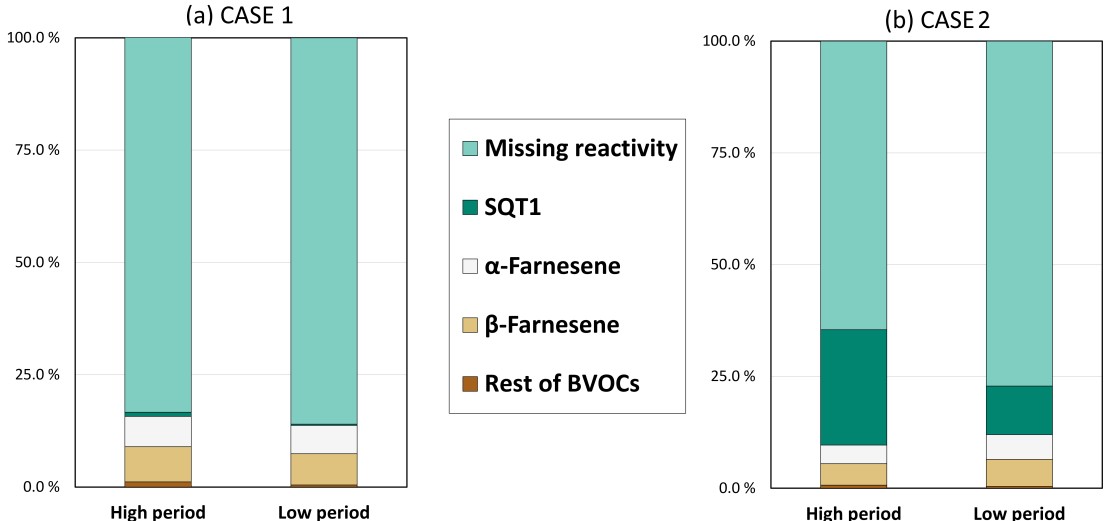

**Figure 7.** Fraction of various contributions of observed BVOCs and missing fraction to the total ozone reactivity for two different cases. (a) Case 1 was calculated after applying the lower reaction rate to the unknown sesquiterpene (SQT1). (b) Case 2 was calculated after applying the higher reaction rate to the unknown sesquiterpene (SQT1).

reactivity period in case 2. It should be noted that both calculated and measured total ozone reactivity values during the low reactivity period is close to or below the detection limit of the instrument and therefore the missing fraction may not be high during the low reactivity period.

During the period of high ozone reactivity, SQTs were the primary contributor to the known fraction of total ozone reactivity, accounting for 16% in case 1 and 35% in case 2. Among the SQTs, the highest contributions were observed from $\beta$-farnesene (8%) in case 1 and an unknown sesquiterpene (26%) in case 2. However, it is important to note that the missing fraction of reactivity was as high as 95% on August 29th, with a mean of 65% during the high reactivity period. These findings highlight the need for further research to identify the unknown compounds responsible for the missing fraction. High fractions of missing OH reactivity have been observed from other studies too especially during higher temperature or drought periods (Nölscher et al., 2013; Praplan et al., 2020). In our study too, the possibility for the existence of undetected compounds that could contribute to reactivity cannot be neglected. GLVs and homoterpenes are compounds with C-C double bonds that can react with ozone. Several studies have highlighted the importance of ozone-GLV reactions (Hamilton et al., 2009; Jain et al., 2014; Barbosa et al., 2017). GLVs are compounds that are released following tissue damage in plants. Besides damage from insects, high temperatures have also found to damage membranes of needles. Kleist et al. (2012) found increase in BVOC emissions when Norway spruce was subjected to high temperatures which was 35 °C. The authors mentioned a thermal threshold that exists beyond which the membrane gets damaged. Pikkarainen et al. (2022) reported that over two growing seasons, the mean emission rate of GLVs from Norway spruce seedlings increased by 350% when the temperature was +4 °C greater than ambient. Most GLVs have reaction rates with $O_3$ in the order $10^{-17}$ - $10^{-16}$ $cm^3\,s^{-1}$ and if emitted considerably these





GLVs can contribute to reactivity. Homoterpenes such as (*E*)-4,8-dimethyl-1,3,7-nonatriene and (E,E)-4,8,12-trimethyltrideca-
1,3,7,11-tetraene are commonly emitted stress BVOCs that are known to be readily oxidised by ozone (Pinto et al., 2007;
Blande et al., 2010). Although both GLVs and homoterpenes can contribute to total ozone reactivity, their emissions should
have been large to contribute to the missing fraction. If emissions were large, they would have been detected by the GC-MS
used in this study. Therefore, it is unlikely that these compounds were responsible for the missing reactivity observed during the
high reactivity period. It is possible that the missing reactivity was caused by entirely unknown compounds or a large number
of compounds including GLVs and homoterpenes, emitted in very low quantities. Future studies could aim to identify these
unknown compounds and also use alternate sampling methods and analytical tools to investigate the potential contribution to
total ozone reactivity.

### 3.5 Temperature dependence of Total Ozone Reactivity

The temperature dependence of total ozone reactivity was also studied. In reference to the temperature dependence of MT
emission from Guenther et al. (1993), a regression line fitted to an exponential curve was calculated for total ozone reactivity
data.

$$R_{O_3,\text{spruce}} = R_{O_3}(T_s) \cdot \exp^{\beta(T - T_s)} \tag{9}$$

where $R_{O_3}(T_s)$ is the normalised emission rate at $T_s$=303.15 K, and $\beta$ is the temperature sensitivity of the $R_{O_3,\text{spruce}}$.

The correlation coefficient for the fit is slightly higher during the low reactivity period ($R^2$ =0.47), suggesting that temper-
ature was the primary driver of reactivity during this time. Reactivity increased gradually with temperature, with most of the
reactivity measured between 280 and 290 K. During the high reactivity period, the calculated $R^2$ value of 0.36 suggests that fac-
tors other than temperature were also driving the reactivity. This indicates that other stress factors may have contributed to the
increase in reactivity during this period. Unlike during the low reactivity period, the data did not follow an exponential curve,
and high reactivity was also observed at lower temperatures. Exposure to high temperatures may have caused tissue damage
(Kleist et al., 2012) and subsequent prolonged emissions of highly reactive compounds, leading to the observed increase in
total ozone reactivity even at lower temperatures.

The $\beta$ depends on the variation of the composition and quantities of ozone-reactive BVOCs in association with temperature.
$\beta$ values from this study are 0.07 K$^{-1}$ and 0.09 K$^{-1}$ for high and low reactivity period respectively. The lower $\beta$ for the
high reactivity period combined with a lesser $R^2$ could be an indication of stress-related non-terpene emissions. The $\beta$ values
for BVOC emissions from Norway spruce studies ranged from 0.008 to 0.3 depending on the compound (Hakola et al., 2017;
Filella et al., 2007; Bourtsoukidis et al., 2013, 2014). The $\beta$ obtained here are closer to the $\beta$ of MTs (0.1 K$^{-1}$) as recommended
by Guenther et al. (2012). SQTs, owing to their higher vapour pressures, have been found to have a stronger temperature
dependence and therefore higher $\beta$ values. However, in a review by Duhl et al. (2008), $\beta$ as low as 0.05 K$^{-1}$ is mentioned.
Hakola et al. (2017) reported $\beta$ values that ranged from 0.02 - 0.06 K$^{-1}$ for SQT emissions from Norway spruce in late
summer. The $\beta$ observed in our study do fall within the range observed from other studies. In our study, the $\beta$ value during
one low reactivity period (17.08 - 27.08) was the highest at 0.13 K$^{-1}$, which lies in between the values of MTs (0.1 K$^{-1}$) and



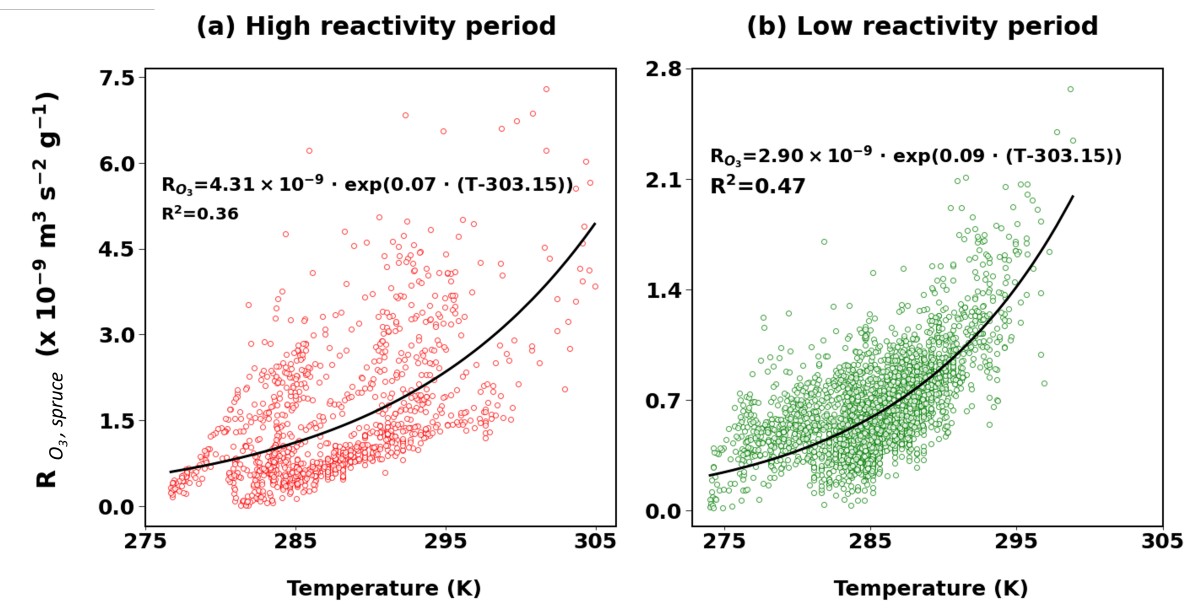

**Figure 8.** Measured total ozone reactivity as a function of temperature for different periods

SQTs (0.17 K$^{-1}$) given in Guenther et al. (2012). Emissions measured during this time were dominated by $\alpha$-farnesene and $\beta$-farnesene. However, the calculated reactivity does not explain reactivity from these two compounds and may indicate the presence of other compounds.

## 4 Conclusions

This study presents the total ozone reactivity measurements conducted on a Norway spruce tree branch in Hyytiälä. Total ozone reactivity was measured directly by the total ozone reactivity monitor (TORM) developed in Finnish Meteorological Institute based on the work by Helmig et al. (2022). These measurements were compared to the calculated ozone reactivities using direct measurements of BVOC emissions from the Norway spruce by the GC-MS.

BVOC emissions from Norway spruce were dominated by the SQTs namely $\beta$-farnesene and $\alpha$-farnesene. Emissions of $\alpha$-farnesene and also *cis*-3-hexenol increased by up to 3 times after the overcast conditions (14.08 - 28.08). Other studies have also observed similar dominance by SQTs from Norway spruce emissions during late summer which were related to stress. An unidentified sesquiterpene was found to be emitted in similar quantities to $\alpha$-farnesene but did not increase as much during the later stage of the campaign. These SQTs correlated poorly with the temperature and PAR and since there were no visible signs of stress, this characteristic emission pattern could be a systemic defence response taken by the tree as suspected in Hakola et al. (2017).



The total ozone reactivity monitor measured a maximum reactivity that is equivalent to 64 µg g$^{-1}$ h$^{-1}$ of $\alpha$-pinene or 0.8 µg g$^{-1}$ h$^{-1}$ of $\beta$-caryophyllene emissions from the measured branch of Norway spruce tree. While high emissions of these compounds can be seen in branch enclosure studies, their concentrations in the atmosphere will be low as they react rapidly with other trace gases in the atmosphere. The SQTs accounted for 13.8% - 35.2% of the measured total ozone reactivity in the low period in case 1 and the high period in case 2. While we have quantified and observed SQTs from a branch of a spruce tree to significantly influence ozone chemistry, a large chunk of total ozone reactivity remains to be explained even during the high reactivity period (65%).

High fraction of missing reactivity was observed when the branch was exposed to heat stress. Spikes in temperature inside the chamber may have induced emissions of reactive compounds undetected by the GC-MS. The measured total ozone reactivity had a temperature dependence when comparing it to the pool emission (temperature dependent) algorithm of vegetation and high reactivity was observed even at lower temperatures. This suggests that stress-related compounds observed during the high reactivity period may have had prolonged emissions due to possible tissue damage caused by high temperatures, and the lower $\beta$ value also suggests the presence of non-terpene compounds induced by stress. As for the diel variation in the total ozone reactivity, a sudden spike at 16:00 coincides with the peak in temperature. The absence of a spike in the calculated ozone reactivities at this hour indicates again the presence of undetected reactive compounds. Observation across different seasons and different trees must be conducted in the future to verify the findings and to provide more insight into stress-related ozone reactivity.

Our study has been able to emphasize the presence of undetected BVOCs from the branch of a Norway spruce that were mostly emitted in times of stress. This also puts an emphasis on the need to expand our search for other classes of compounds than the ones that are often measured. Stress episodes in the form of both biotic and abiotic have become increasingly common, especially in the Arctic, and these events will cause the release of a vast range of BVOCs not known earlier.

*Data availability.* The data are available from https://doi.org/10.23728/b2share.429af0b834a54a2e824935fb3b1ea518. The raw data measured by TORM, GC-MS and other sensors will be made available upon request. The ambient data was taken from FMI open weather data (URL: https://en.ilmatieteenlaitos.fi/download-observations; last accessed on 13-07-2022).



## Appendix A: Biogenic Volatile Organic Compound (BVOCs) detected using GC-MS.

**Table A1.** Mean emission rates ($\mathrm{ng\,g^{-1}\,h^{-1}}$) and ozone reactivity ($\mathrm{m^3\,s^{-2}\,g^{-1}}$) of different compounds observe from Norway spruce. Two ozone reactivity values are calculated for SQT1 that correspond to case 1 and case 2 respectively.

| Compound | Emission rate | Ozone reactivity |
|---|---|---|
| Methacrolein | 0.0005 | $1.6\times 10^{-18}$ |
| MBO | 1.3 | $2.6\times 10^{-14}$ |
| *cis*-3-Hexenol | 2.2 | $2.4\times 10^{-13}$ |
| Isoprene | 0.4 | $6.8\times 10^{-15}$ |
| **Monoterpenes** | | |
| $\alpha$-Pinene | 2.9 | $3.5\times 10^{-13}$ |
| Camphene | 0.9 | $5.9\times 10^{-16}$ |
| Myrcene | 2.1 | $1.2\times 10^{-12}$ |
| $\beta$-Pinene | 0.4 | $8.9\times 10^{-15}$ |
| Carene | 0.4 | $2.4\times 10^{-14}$ |
| *p*-Cymene | 0.1 | $3.2\times 10^{-18}$ |
| Limonene | 1.7 | $4.7\times 10^{-13}$ |
| Terpinolene | 0.1 | $1.1\times 10^{-13}$ |
| Sum MTs | 8.6 | |
| **Oxygenated monoterpenes** | | |
| 1,8-Cineol | 1.5 | $2.5\times 10^{-16}$ |
| Linalool | 0.8 | $4.0\times 10^{-13}$ |
| Bornylacetate | 0.4 | $2.2\times 10^{-17}$ |
| Sum OMTs | 2.7 | |
| **Sesquiterpenes** | | |
| $\beta$-Farnesene | 137.5 | $6.5\times 10^{-11}$ |
| $\beta$-Caryophyllene | 0.5 | $5.0\times 10^{-12}$ |
| $\alpha$-Humulene | 0.005 | $4.6\times 10^{-14}$ |
| $\alpha$-Farnesene | 112.4 | $5.5\times 10^{-11}$ |
| Unknown Sesquiterpene (SQT1) | 30.6 | $5.1\times 10^{-12}$ / $2.3\times 10^{-10}$ |
| Sum SQTs | 281 | |





# Appendix B: Total ozone reactivity monitor

## B1 Calibration of TORM

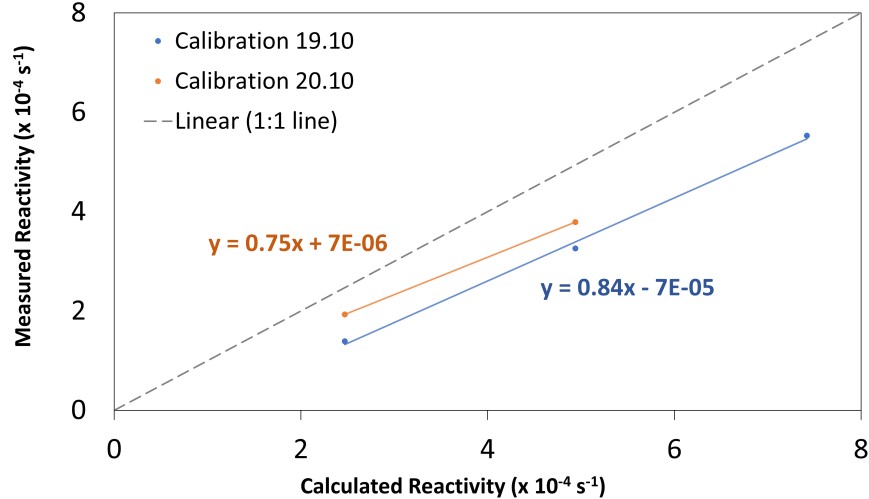

**Figure B1.** Calibration of the total ozone reactivity monitor. The mean slope and intercept from the two calibration lines were used as the correction factors.



## B2 Correction of measured reactivity

The reaction between ozone and a fast reacting compound was modelled using the pykpp chemical box model in python programming language. The pykpp (https://github.com/barronh/pykpp/, last accessed on 14 March 2023) is based on KPP (the Kinetics Pre-Processor; Damian et al., 2002) which generates a box model that can be run in python programming language. The model used chemical equations of the form VOC + $O_3 \rightarrow$ Dummy to describe the chemical degradation of VOCs in the presence of ozone. The reaction rates for the reaction between different VOCs and ozone are given in Table 2. The model

considered 1 ppb of a compound with a fast reaction rate ($1.2 \times 10^{-14}$ cm$^3$ s$^{-1}$), 100 ppb of ozone at 298 K and 1 atm. The ozone decay under these conditions have been shown in fig. B2.

The total ozone reactivity calculated using eqn. 4 will not be the same between any two points in time in fig. B2 due to the exponential decay of ozone in the presence of a fast reacting compound. However, ozone decay is linear for a short time interval ($\Delta t_s$) and this is the true total ozone reactivity in the presence of a fast reacting compound. The residence time in

TORM is considered to be 108 seconds and by which the total ozone reactivity will be underestimated in these conditions. The correction factor to calculate the true total ozone reactivity can be obtained in the following way considering $\Delta t_s$ = 41.9 seconds, $\Delta [O_3]_s$ = 99.35 ppb, $\Delta t_l$ = 108.7 seconds , $\Delta [O_3]_l$=99.04 ppb and $[O_3]_o$=100 ppb.

Let reactivity calculated for the short time ($\Delta t_s$) be:

$$R_{O_3,s} = \frac{\Delta [O_3]_s}{[O_3]_o \Delta t_s} \tag{B1}$$

And let the reactivity calculated for the longer time ($\Delta t_s$) be:

$$R_{O_3,l} = \frac{\Delta [O_3]_l}{[O_3]_o \Delta t_l} \tag{B2}$$

Then the correction factor for totol ozone reactivity for a fast reacting compound is

$$\frac{R_{O_3,s}}{R_{O_3,l}} = \frac{\Delta [O_3]_s \Delta t_l}{\Delta [O_3]_l \Delta t_s} = 1.7 \tag{B3}$$



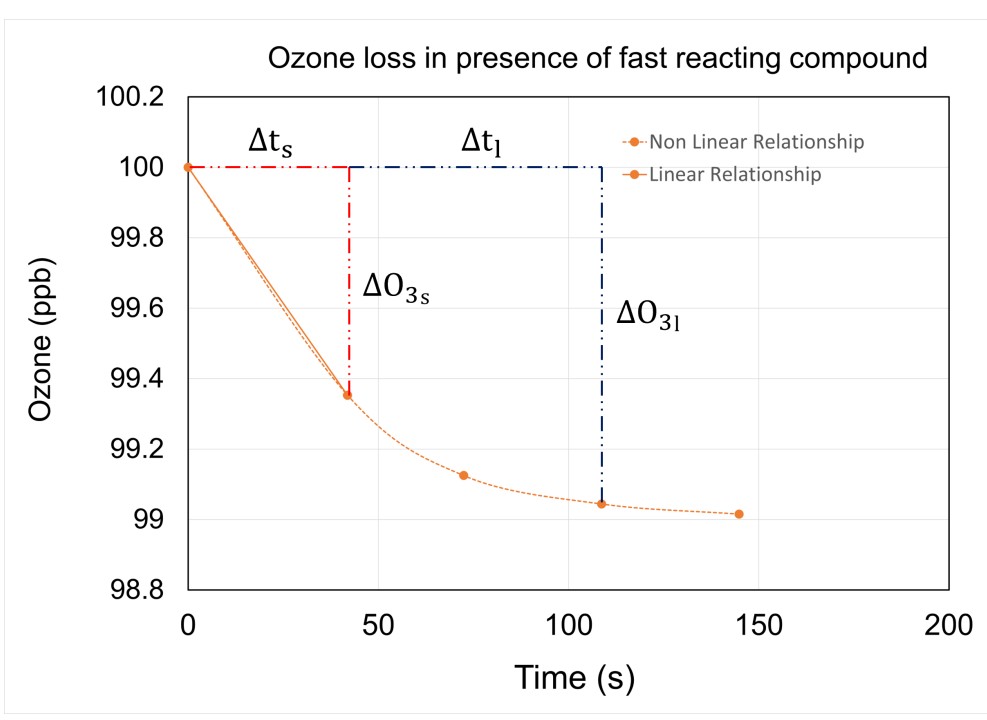

**Figure B2.** Ozone loss modelled using pyKPP in the presence of 1 ppb of a fast reacting compound and 100 ppb of ozone.



*Author contributions.* SJT conducted the ozone reactivity measurements, performed the data analysis and led the writing of the manuscript.

TT and HH conducted the GC-MS measurements, analysed the data produced by the GC-MS and commented on the manuscript. APP was the

principal investigator, designed the measurement campaign, assisted in the data analysis of ozone reactivity measurements and commented

on the manuscript. FB contributed to the discussions of the results and provided input for the manuscript.

*Competing interests.* The authors declare that they have no conflict of interest.

*Acknowledgements.* The research was supported by the Academy of Finland (decisions nos. 307797 and 335319). We also thank SMEAR II

station and their staff for providing infrastructure and support during the campaign.



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
