# Peer review of "Undetected BVOCs from Norway spruce drive total ozone reactivity measurements"

_EGUsphere, 2023_

## Author Comment (AC1)

**Undetected BVOCs from Norway spruce drive total ozone reactivity measurements**

Steven Job Thomas[1,2], Toni Tykkä[1], Heidi Hellén[1], Federico Bianchi[2], and Arnaud P. Praplan[1]

[1]Atmospheric Composition Research, Finnish Meteorological Institute, Helsinki, 00101, Finland.
[2]Institute for Atmospheric and Earth System Research/Physics, Faculty of Science, University of Helsinki, Helsinki, 00560, Finland.

**Correspondence:** Steven Job Thomas (steven.thomas@helsinki.fi)

**Reviewer 1 comments and response**

**We thank the reviewer for the valuable comments which have helped us improve the manuscript. We provide here answers and mention the changes that have been made to address the referee's concerns. The referee's comments are in normal font while the replies are given in bold and blue text. We would like to draw your attention to the fact that we have introduced new nomenclature: the normalised total ozone reactivity is now referred to as TOZRE (total ozone reactivity of emissions), and the normalised calculated ozone reactivity is termed COZRE (calculated ozone reactivity of emissions). This alteration has been introduced to mitigate potential confusion, given that the normalised total ozone reactivity is denoted in units of m$^3$ s$^{-2}$ g$^{-1}$, while reactivity is inherently expressed in units of s$^{-1}$.**

A. General comments:

In this manuscript, BVOCs emission from Norway spruce was explored utilizing the Total Ozone Reactivity Monitor (TORM) which has been established recently based on the differential O3 monitor (Helmig et al., 2022). This study is positioned as a demonstrative research of the TORM instrument. Samples were prepared by use of the branch enclosure. Simultaneously, various BVOC species were individually monitored by the cold trap / thermal desorption / GC-MS technique which could quantify many ozone-reactive species including representative sesquiterpenes. BVOCs emission was characterized (eg. temperature dependence). Significant gap (missing reactivity) between observation by TORM and theoretical calculation based on GC-MS analysis was confirmed, suggesting experimentally that further exploration on unmeasured BVOC species could be important. Especially, missing ozone reactivity remained still significant although authors monitored representative ozone-reactive sesquiterpenes (eg. b-caryophyllene) and discussed on the possibility of the contribution by some unmeasured species like GLVs.

Totally, the reviewer believes that this work can be a breakthrough of O3-BVOCs chemistry in the atmosphere and has an important implication and is significant enough to be published in this journal. However, the present manuscript leaves several minor points to be clarified, in order for readers to understand descriptions and to recognize the significance and descriptions of this study more clearly. Especially, descriptions on the assumption of linear ozone decay in the reactor of TORM can be confusing for the quantitative discussion in this study. Minor revision is necessary.

B. Specific comments:

B1) Line 110 and around:

 The descriptions on principles of the TORM instrument are one of the most important points in order to evaluate and understand the results of this study properly and quantitatively. Especially, the assumption of linear ozone decay in the reactor of TORM can be directly linked to invalid quantification of RO3 as suggested around Lines 220-221 and Section B2. Therefore, minimum descriptions on the principle should be added (including strict equation for exponential decay, linear assumption, and notice on fast reacting compounds). For example, …

… Strictly, RO3 can be determined from exponential decay of ozone in the reactor:

RO3 = - {LN(O3(t) / O3(0) ) } / Dt.

In this study, … (original Line 119)

 RO3 ~ (original Eq.2) .

… This assumption can be valid when fast reacting compounds are negligible and linear decay of ozone is proper. …

**The statement below has been incorporated into the revised manuscript at line 112:**

**Using this equation, $R_{O3}$ cannot be calculated at all situations. This equation becomes invalid in the presence of a fast-reacting compound where ozone decays exponentially and breaks the assumption of pseudo-first order reaction upon which this equation was derived (Helmig et al., 2022).**

B2) Line 114 and around:

 Would you please clarify the model number and supplier of each O3 analyzer?

**Both ozone analysers used in the study were of the same model, specifically Model 49i from Thermo Scientific in Waltham, MA, USA.**

**The statement below has been incorporated into the revised manuscript at lines 118 and 121:**

**Model 49i, Thermo Scientific in Waltham, MA, USA.**

B3) Lines 131 - 135 and Figure B1:

 In the present preprint, the readers can recognize only the fact that, as a result of calibration, the TORM underestimated the standard concentrations of 4 BVOCs mixture. Why does such an underestimation occur? Why is the correction described as Eq.5? Is such a correction proper quantitatively? Would you please indicate authors' ideas at least in the manuscript.

 Relatedly, is the prepared standard sample (mixture of 4 BVOCs compounds, 200 ppbv each)

proper as the conditions/settings of calibration during measurements in this study? Please clarify authors' ideas in the text. Additionally, is the standard sample valid in view of the linear assumption in Eq.2?

One of the major reasons behind the underestimation of total reactivity measured using TORM is due to the constant addition of ozone into the reactor which hinders the reaction with the BVOCs with the constant addition of ozone. This causes the differential term in the numerator of equation (2), to be underestimated. This in turn causes the reactivity to be underestimated from the calculated reactivity.

The correction factor's equation, utilized to address these discrepancies, is obtained from the graph itself, and a detailed derivation of this equation will be included. For calibration purposes, a standard mixture was obtained from the National Physical Laboratory, and the calibration procedure was promptly conducted after the measurement period. Throughout this process, the instrument was securely housed in an air-conditioned container, ensuring consistent conditions within. As no alterations were made to the instrument or its connections, except for connecting the calibration gas cylinder, we assume the instrument conditions remained unchanged from the measurement period.

The standard sample employed in calibration contained only four BVOCs, none of which are fast-reacting compounds. Consequently, in the presence of these BVOCs, ozone decay followed a linear pattern considering the residence time of the reactor used in our study (108 seconds). This calibration approach with non-reactive BVOCs allowed us to account for potential deviations in the total ozone reactivity measurements, improving the accuracy and reliability of our results.

The derivation of the factors has been given in the appendix of the revised manuscript and the statement below has been incorporated into the revised manuscript at line 138:

The continuous injection of ozone into the reactor counteracts the ozone decay resulting from BVOCs, which leads to an underestimation of reactivity. To address this issue, TORM needs to be calibrated with measured reactivity using a standard.

B4) After Line 145:

NO rapidly reacts with ozone and can interfere total ozone reactivity measurements. Would you add descriptions on typical concentrations of residual NO in zero air and samples? And, can such NO interference be removed or reduced in Eq.3?

NO interference can be removed by subtracting NO concentration from the delta O3 signal. NO concentration measurements were not carried out during the campaign; nevertheless, ambient NO concentrations from Hyytiälä data indicated that NO concentrations were generally below 0.1 ppb throughout most of the campaign. Had there been substantial amounts of NO present inside the enclosure or during zero air measurements, this would have been evident in the total ozone reactivity measurements. However, the stable total ozone reactivity values obtained during the zero air measurements suggest that there was no significant interference from NO. If at all, the zero air was contaminated with NO, this would be considered when subtracting the background measurements.

**In conclusion, despite not directly measuring NO concentrations during the campaign, the stability of total ozone reactivity values during zero air measurements provides strong evidence that NO did not substantially affect the measurements.**

B5)Lines 219-224, Figure B2, Lines 344-363 and around:

In these paragraphs and figure, to evaluate the upper limit of unknown SQTs' contribution to ozone reactivity (RO3) as 'case2', correction in RO3 measurement for fast reacting compounds is explained.

1) Around line 220, such a purpose of these descriptions should be clarified by adding a little description on 'case2'.

**Case 1 and Case 2 represent two different scenarios for calculating reactivity, involving distinct reaction rates for the unknown sesquiterpene (SQT1), as previously mentioned in lines 213-218. In Case 2, a faster reaction rate is applied to SQT1, classifying it as a fast-reacting compound. As we have pointed out, the presence of fast-reacting compounds can lead to an underestimation of the measured reactivity.**

**To address this issue, we introduced a correction to the measured reactivity, as already indicated from line 222 and in Appendix B2. The correction factor is applied to the measured reactivity in order to account for the influence of SQT1, specifically when the faster reaction rate is used. Appendix B2 elaborates on this correction process, allowing for a meaningful comparison between Case 2 and the measured reactivity.**

**By applying the correction, we ensure that the measured reactivity tries to accurately represents the true reactivity, even in the presence of fast-reacting compounds like SQT1 (in case 2). We hope that this additional clarification resolves any confusion and emphasizes the importance of the correction in considering the impact of fast-reacting compounds on the measured reactivity.**

**Changes have been made to the second paragraph under section 3.3. The following is the paragraph after the changes.**

**To compare TOZRE with COZRE, we considered two different cases of COZREs. In each case, the unknown sesquiterpene (SQT1) was assigned a different reaction rate with ozone before determining the COZRE. In case 1, the reaction rate of cedrene which is on the lower end of the known sesquiterpene reaction rates, was assigned (2.2 x 10$^{-16}$ cm$^3$ s$^{-1}$). While in case 2, we employed a faster reaction rate with ozone (1.2 x 10$^{-14}$cm$^3$ s$^{-1}$), which is equivalent to the reaction rate of β-caryophyllene. A reaction rate faster than this would lead to COZRE surpassing TOZRE, which is not possible. The two reaction rates will explain how COZRE values compare with TOZRE. After comparing the TORM signal with the two cases, any unexplained signal confirms the presence of unmeasured compounds.**

**The statement "(as SQT1 is a fast-reacting compound in case 2)" has been added into the revised manuscript at line 241:**

2) In Appendix B2, such a purpose of these descriptions should be clarified for readers to avoid

confusing between underestimations in B1 and those in B2. For example:

Title of B2: Correction of measured reactivity for fast reacting compounds in 'case2'

**DONE**

**The title for appendix B2 was changed to "Correction of measured reactivity in the presence of fast reacting compound (required to compare measured reactivity with Case 2)."**

3) In case except for 'case2', is such a correction not necessary? When is such a correction necessary? Would you indicate authors' ideas at least in section B2?

**Total ozone reactivity measured by TORM encounters limitations when a fast-reacting compound is present in the sample. The pseudo-first-order assumption, upon which Equation 2 is derived, becomes invalid under such circumstances, as described in Helmig et al. (2022). The presence of a fast-reacting compound leads to an exponential decay of ozone, resulting in an underestimation of total ozone reactivity by TORM. To address this issue, we have introduced a correction factor obtained from Appendix B2. This correction factor is applied to the measured reactivity whenever emissions of SQT1 are observed beyond 0.2 ppb.**

**We have given the reasons for applying the correction factor and specified when it should be implemented. The descriptions are given from line 230 to line 235.**

**An additional paragraph has been added to Appendix B2 to introduce the reasons for the correction:**

**When a fast-reacting compound is present in the sample air, the total ozone reactivity, determined using Eq. 2 tends to be underestimated. This is due to the exponential decay of ozone, which violates the assumption of a pseudo-first order reaction, upon which this equation was derived (Helmig et al., 2022). In our case 2, SQT1 is assumed to be a fast-reacting compound. To facilitate a comparison between the total ozone reactivity and the calculated reactivity, a correction must be applied to account for the underestimated total ozone reactivity. The correction factor obtained in this section was applied to 33% of the TORM dataset, specifically when emissions of SQT1 were observed to increase beyond 0.2 ppb.**

4) From the value of k (Line 350), authors may use k of b-caryophyllene as the upper limit. Would you indicate such authors' ideas at least in section B2?

**We took a careful approach to address the correction of measured reactivity in the presence of the fast-reacting compound, SQT1. To calculate the reactivity for Case 2 from the GC-MS measurements, we assigned the reaction rate of β-caryophyllene ($1.2 \times 10^{-14}$ cm$^3$ s$^{-1}$), known for its high reactivity with ozone, to SQT1. Our calculations indicated that any reaction rate value exceeding this led to an overestimation of the calculated reactivity, surpassing the measured reactivity, which is physically implausible.**

**Considering this, we assumed that the observed SQT1 compound could not possess a reaction rate beyond this value. Thus, we adopted the reaction rate of β-caryophyllene for Case 2's**

**calculated reactivity as the upper limit. Consequently, we utilized the same reaction rate for the correction of measured reactivity in the presence of the fast-reacting compound, SQT1.**

5) Is it not difficult in principle to evaluate the strict RO3 values based on exponential decay of ozone from observed values as follows?

$$O3(t) / O3(0) = exp \{ - RO3 \; Dt \}$$

$$O3(t) = O3(0) - DO3$$

$$=>=> RO3 \; - \{ LN (1 - DO3/O3(0) ) \} / Dt$$

How about results in this study? Can any consideration be possible by use of strictly evaluated RO3? If it is not possible, please show us the reasons. And would you please indicate such situations in the text as possible?

**The equation used for total ozone reactivity is derived from an approximate solution obtained through Taylor series expansion, as detailed in Helmig et al., (2022). The specific equation RO3 = - { LN (1 - DO3/O3(0) ) } / Dt is a preliminary step before applying the Taylor series expansion, and it is also based on the pseudo first-order kinetics. The measured reactivity from equation 2 in the manuscript and from this equation gives the same result.**

B6) Line 273 and around:

There is a leap in logic around 'Therefore'. For example, please clarify as follows:

… However, these compounds were insignificantly detected by GC-MS. Therefore, …

**GLVs are common compounds that are emitted upon stress. If emissions of the GLVs were large, they would have contributed to the high ozone reactivity detected by torm. But if GLV emissions were high, the GC-MS used for the measurements would have detected them, which is not the case. Our GC-MS could not detect those which means, GLV emissions may have not been high enough. Therefore, we might be able to rule out the possibility that GLVs could have contributed to the missing reactivity.**

**The line has been changed to**

**…However, although both GLVs and homoterpenes can contribute to total ozone reactivity, their emissions were below the detection limits of the GC-MS used in this study. Therefore, it is unlikely that these compounds were solely responsible for the missing reactivity observed during the high reactivity period….**

B7) Figure 8:

If the green points (or their regression line) in Fig.8b are plotted simultaneously in Fig8a, do the lower envelope of red points overlap on the green points (or their regression line) ? If so, can the new figure (simultaneous plots of red and green points) be added in the appendix of the manuscript in order to clarify the descriptions in Lines 284-304 (red points = temperature dependence of MT

(green points) + spikes due to stress) ? Alternatively, would you show such an additional plot in the interactive public discussion?

**Thank you for giving us this suggestion. Figure 8 in the manuscript has been replaced with the figure seen below.**

[Figure]

Fig 1: TOZRE as a function of temperature for high reactivity period (red dots) and low reactivity period (green dots). Red and green line indicate the exponential fit for the high and low reactivity periods respectively.

C. Technical corrections:

C1) Table 1, Figure 3, Figure 7, Figure 8:

Please clarify in the text where these table and figures are explained and/or referred.

**DONE.**

End of Comments.

**Citation**: https://doi.org/10.5194/egusphere-2023-839-RC1

**Undetected BVOCs from Norway spruce drive total ozone reactivity measurements**

Steven Job Thomas[1,2], Toni Tykkä[1], Heidi Hellén[1], Federico Bianchi[2], and Arnaud P. Praplan[1]

[1]Atmospheric Composition Research, Finnish Meteorological Institute, Helsinki, 00101, Finland.
[2]Institute for Atmospheric and Earth System Research/Physics, Faculty of Science, University of Helsinki, Helsinki, 00560, Finland.

**Correspondence:** Steven Job Thomas (steven.thomas@helsinki.fi)

**Reviewer 2 comments and response**

**We thank the reviewer for the valuable comments which have helped us improve the manuscript. We provide here answers and mention the changes that have been made to address the referee's concerns. The referee's comments are in normal font while the replies are given in bold and blue text.**

The manuscript "Undetected BVOCs from Norway spruce drive total ozone reactivity measurements" by Thomas et al. presents a set of measurements of total ozone reactivity in a branch enclosure of a spruce tree and compares these measurements with calculations using observations of BVOC by gas-chromatography. The manuscript is a nice demonstration of the potential use of ozone reactivity measurements and it is within the scope of the journal. I have a few comments that need to be addressed, but otherwise I think the manuscript is suitable for publication.

Main Comments
* * *
1. The unit m3 s-2 g-1 is used for ozone reactivity throughout the manuscript. I think this is not correct because reactivity is, by definition, just the inverse of the chemical lifetime and should therefore be in s-1. The authors themselves define it as such with Equation 1. I assume that the reactivity presented in the manuscript is weighted by mass of the branch and scaled by time in order to be in m3 s-2 g-1, but this is not mentioned anywhere. I would suggest that, first of all, this information is added. And, second, that instead of "ozone reactivity" the authors use a different term, where appropriate.

**While the unit of reactivity is indeed s$^{-1}$, the total ozone reactivity in this paper has been normalised by the mass of the branch and the flow of air through the enclosure. The normalisation has been explained in section 2.2 and section 2.3 for the measured and calculated ozone reactivities respectively. Changes have been brought throughout the paper to avoid any confusion.**

**Total ozone reactivity of emissions measured by TORM has been called as TOZRE while calculated ozone reactivity of emissions has been termed as COZRE.**

In section 2.1 the authors mention that PAR measurements after August 23 were biased due to the shade of a nearby scaffolding. PAR is used in section 3.2 to discuss the emissions of some BVOCs. I would expect that only the data prior to August 23 are considered in figure 4 (and related discussion in section 3.2). Please clarify that this is the case, and if not, why unrepresentative PAR measurements have been used.

**Thank you for your valuable comments and observations. We are fully aware of the potential bias introduced by placing the PAR sensor behind the scaffolding, leading to shaded conditions. However, it is important to note that the chamber itself was also mostly under shade throughout the day as it was one of the lowest branches on the tree. Therefore, we believe, the PAR sensor might have missed direct sunlight only during specific hours, as evident from the dip observed in Fig 4(b) after noon.**

**Despite this limitation, we have thoroughly examined the diel pattern of PAR and discovered that it still exhibits an almost consistent alignment with temperature variations, indicating a certain level of data reliability. Since no major analyses or conclusions rely solely on the PAR measurements, we agree that it may not be necessary to remove the entire dataset.**

**We have already mentioned the issue with PAR in our manuscript. However, the following statement has been added in the revised manuscript in section 3.2 at line 218.**

**However, it is essential to acknowledge that this conclusion cannot be made with absolute confidence, as the PAR measurements during the second half of the campaign may have been underestimated for a couple of hours during the day.**

2. In section 2.3, the quantification of a-Farnesene and of bornyl acetate is determined assuming the same sensitivity as b-caryophyllene and nopinone, respectively. I suggest to add a comment on the potential error introduced by this assumption.

**The detection limits of terpenes vary between 0.08 ng g$^{-1}$ h$^{-1}$ and 1.14 ng g$^{-1}$ h$^{-1}$ and the uncertainty of the measurements lies between 18% - 25%. While the detection limits of MAC, MBO and cis-3-hexenol are 0.02 ng g$^{-1}$ h$^{-1}$, 0.35 ng g$^{-1}$ h$^{-1}$, and 0.47 ng g$^{-1}$ h$^{-1}$ respectively. The uncertainty of compounds lacking authentic standards are higher than the others.**

**Statement added in Section 2.3 In-situ emission measurements:**

**The detection limits for terpenes range from 0.02 ng g$^{-1}$ h$^{-1}$ to 0.41 ng g$^{-1}$ h$^{-1}$, with measurement uncertainties falling within the 18% - 25% range. As for methacrolein, MBO, and *cis*-3-hexenol, their detection limits are 0.03 ng g$^{-1}$ h$^{-1}$, 0.35 ng g$^{-1}$ h$^{-1}$, and 0.48 ng g$^{-1}$ h$^{-1}$, respectively. Compounds lacking authentic standards exhibited higher levels of uncertainty compared to the others.**

3. In section 3.3 (pages 11-12) the discussion focuses on the comparison between observed and measured reactivity. A correction is applied to the observations (please state explicitly) and then the data with and without correction are compared to calculations in case 1 and 2. I have questions regarding this procedure:

a) First, why is the correction only applied to the data compared to case 2? Sesquiterpenes are present also in case 1, the only difference between the cases is that the unidentified SQT1 is

given a slow rate coefficient, but the issue described in figure B2 should be valid for all sesquiterpenes not just SQT1.

**The equation of total ozone reactivity (eqn 2) becomes invalid when a fast-reacting compound is present in the sample air. In the presence of a fast-reacting compound, the ozone decays exponentially which breaks the pseudo first order assumption upon which the equation was derived (Helmig et al., 2022). This causes the total ozone reactivity to be underestimated since the fast-reacting compound finishes very quickly than a slow reacting compound for the same residence time in the reactor. This requires correction.**

**Not all sesquiterpenes classify as 'fact-reacting' compounds. Among the identified BVOCs in our study, only a couple of sesquiterpenes, namely β-caryophyllene and α-humulene react quickly (reaction rate: $1.2 \times 10^{-14}$ $cm^3$ $s^{-1}$) with ozone. In our study we have identified β-caryophyllene and α-humulene but at very minor quantities while other sesquiterpenes (reaction rates that are orders of magnitude lower than that of β-caryophyllene) have been found in much higher quantities. The unknown sesquiterpene (SQT1) was emitted a lot but not observed every day. Being an unknown compound, it's reaction rate with ozone cannot be determined either. Therefore, it is not possible to determine the calculated ozone reactivity. To compare the calculated ozone reactivity with the measured reactivity we came up with two different scenarios. Case 1: Calculated reactivity from identified compounds considering when SQT1 is a slow-reacting compound and Case 2: Calculated reactivity from identified compounds considering when SQT1 is a fast-reacting compound. The two cases of calculated reactivities were well within the measured reactivity making a possibility that the reaction rate of SQT1 with ozone could lie anywhere within this range of assumed minimum and maximum or even lower. Now in both cases there is β-caryophyllene and α-humulene but in very less quantities (so that it won't affect significantly the TORM dO3 signal) so the measured reactivity will not be affected by this. And since we consider SQT1 to be a slow reacting compound in case 1 of calculated reactivity, the comparison between case 1 and measured reactivity is straightforward without any correction needed for the measured reactivity. However, in Case 2 of calculated reactivity, SQT1 is a fast-reacting compound and since it was emitted in large quantities on several days, the measured reactivity might have been underestimated on those days when SQT1 was emitted (as equation for TORM becomes invalid for fast reacting compounds). To account for the underestimation of measured reactivity caused by SQT1, a correction factor must be applied which was done using the pykpp Box Model. The correction factor was applied to the measured reactivity during days when SQT1 was emitted beyond the detection limit of TORM. The correction factor will change based on the reaction rate of the compound and therefore the compound itself. The correction must be done every time a fast-reacting compound is detected. Besides identifying a fast-reacting compound with the help of conventional instruments, we cannot omit the fact that a fast-reacting compound could be present in the sample air that could not be detected. This will cause interferences in the measurement but will be a part of the missing reactivity.**

b) Second, if the correction is calculated with a box model (as described in appendix B2) to account for the differentials in BVOC ozone reactivities, I would expect it to be dependent on the particular mixture of BVOC, which means both the species in the mixture and their concentrations. I would therefore expect the result of the model calculation to be different for each data point of the campaign. Instead, from equation B3 it looks like a constant correction factor was applied. The

authors should explain this point better, because it hugely impacts both the interpretation of figure 6 and discussion on missing reactivity, and therefore the overall conclusions of the paper.

[Figure]

Fig 1: Ozone decay inside the reactor modelled for two scenarios.

**Correction factor when only SQT1 is considered: 1.7 (from the manuscript)**

**Derivation of correction factor when all compounds are considered:**

**Concentration of all compounds given in the model are the mean concentrations of compounds observed in our study.**

**$t_s$ = 36.9 s, $t_l$ = 109.64 s, $[O_3]_o$ = 100 ppb, $\Delta[O_3]_s$ = 1.2   $\Delta[O_3]_l$ = 2.2**

**$\dfrac{R_{O3,s}}{R_{O3,l}}$ = 1.6**

**Upon careful investigation, we observed that the correction factor (the ratio) remains relatively consistent across different scenarios. We have included an example graph and provided relevant calculations to illustrate this point. Regardless of whether all compounds or only SQT1 were considered, the corrections applied to the measured reactivity did not vary significantly. Although we noticed a larger magnitude of ozone depletion when all compounds were included, the ratio of reactivity under the pseudo-first-order assumption ($R_{O3,s}$) to reactivity at exponential decay ($R_{O3,l}$) showed minimal variation between these two scenarios. This behaviour can be attributed to the rapid consumption of SQT1 by ozone inside the reactor, leading to it's depletion, while other compounds continue to be consumed at a slower rate. In reality, SQT1 was emitted in considerable quantities on days when it was observed, thus dictating the rate at which ozone decays in the system. Consequently, we focused on considering only SQT1 to apply the correction factor to the measured reactivity.**

**Moreover, the correction factor obtained from Appendix B2 was applied to the measured reactivity only when the concentration of the fast-reacting compounds (SQT1) exceeded 0.2 ppb, which is above the detection limit of TORM.**

c) In appendix B1, two calibrations are shown. I would argue that the 20.10 calibration cannot be considered reliable. It has only two points, so the slope is kind of meaningless. I suggest that only the 19.10 calibration should be used in the manuscript and all the numbers updated accordingly.

[Figure]

Fig 2: Calibration curves of TORM.

We appreciate your concern regarding the reliability of the calibration conducted on the 20th of October, which relies on a limited dataset of only two data points. While we acknowledge the potential uncertainties associated with using a small dataset, we find some alignment between the calibration results on both dates. We think although only two points were obtained, the calibration is not entirely meaningless.

By treating the entire dataset as a single entity a linear fit equation was obtained. The attached plot shows the fit (black line) and equation for the whole dataset combined. Interestingly, we observe that this equation aligns closely with the one already utilized in our manuscript. The calibration equation used in our study is derived from the average of the slope and intercept of the calibration conducted on the 19th and 20th. The consistency between the calibration equations reinforces the validity of our approach. Therefore, we think that the calibration data remains meaningful and contributes to the robustness of our results.

Minor Comments

line 3 (and elsewhere in the text): should it be "low volatility"?

The usage of low volatile organic compounds is correct.

line 26: what do you mean with "other emissions"?

…50% of other emissions…mean 50% of other BVOC emissions.

The statement has been changed to

**Based on the latest biogenic emission model, the Model of Emissions of Gases and Aerosols from Nature (MEGAN v2.1; Guenther et al., 2012), it is estimated that BVOCs from tropical regions account for 80% of terpenoid emissions and 50% of other VOC emissions. In contrast, trees from other biomes collectively contribute only 10% of the total BVOC emissions.**

line 36: delete "on".

**The sentence has been changed to**

**Several studies conducted in the boreal forest in Finland (Hyytiälä) have observed monoterpenes such as α-pinene undergoing ozonolysis or OH-initiated reactions, resulting in the formation of highly oxygenated molecules that may lead to the formation of SOA.**

line 234: you should state explicitly that R(O3,corr) was corrected for the underestimation observed during the calibration, so that the procedure is more clear. Also please add the detection limit and uncertainty of all instruments somewhere in this section.

**Error/uncertainty is produced by the uncertainty of the residence time in the reactor (10%), ozone monitor measurements (1%) and the variability in the background measurements (9%). Therefore, the total uncertainty in reactivity measurements comes out to be 20%.**

**The detection limit of the instrument is $4.44364 \times 10^{-5}$ s$^{-1}$.**

**Statement added in Section 2.2.1 Total Ozone Reactivity Monitor (TORM):**

**The overall uncertainty associated with TORM can be evaluated by considering various factors. These include the uncertainty in determining the residence time (estimated at 10% based on Helmig et al., 2022), the uncertainty of the ozone monitors (1%), and the median variability of background measurements (9%), resulting in a combined uncertainty of approximately 20%.**

**Detection limit of GC-MS measurements and uncertainty:**

**Statement added in Section 2.3 In-situ emission measurements:**

**The detection limits for terpenes range from 0.02 ng g$^{-1}$ h$^{-1}$ to 0.41 ng g$^{-1}$ h$^{-1}$, with measurement uncertainties falling within the 18% - 25% range. As for MAC, MBO, and *cis*-3-hexenol, their detection limits are 0.03 ng g$^{-1}$ h$^{-1}$, 0.35 ng g$^{-1}$ h$^{-1}$, and 0.48 ng g$^{-1}$ h$^{-1}$, respectively. Compounds lacking authentic standards exhibited higher levels of uncertainty compared to the others.**

line 189: was SQT1 correlated to any other identified BVOC and/or temperature, PAR or other observed parameters? Were there other compounds that were present but not identified?

**SQT1 had a good correlation with α-farnesene (r=0.65) and β-farnesene (r=0.79). And SQT1 correlated poorly with both temperature, PAR and other observed parameters. We assume that SQT1 is a compound induced from the systemic defence of the tree. There was no visible herbivory damage on the tree.**

**No other unknown compounds were present.**

line 212: add some references for previous enclosure studies.

**DONE**

line 249: change to "very low".

**DONE**

lines 249-250: what does it mean "partly"? By higher rate you mean the faster or the slower rate coefficient? Please be more quantitative and precise.

**DONE**

**The statement has been revised to the following:**

**However, the low reactivity period in September (3.09 - 10.09) could be explained partly (65%) by COZRE when the faster reaction rate was applied (case 2).**

figure 2: relative humidity is clearly high inside the branch enclosure, often around 100%. Can the authors comment on whether the high humidity can affect the determination of ozone by the Differential Monitor?

**We agree that RH levels can potentially impact ozone measurements using the Differential Monitor. To address this concern, our experimental setup includes Nafion dryers within the Total Ozone Reactive Measurement (TORM) system. These dryers play a crucial role in removing moisture from the air, ensuring that the differential monitors record ozone levels in a consistent and accurate manner. Furthermore, we would like to mention that the effect of Nafion dryers on RH levels and their contribution to stabilizing the differential signal has been thoroughly investigated and reported in Helmig et al., 2022. This study provides comprehensive insights into how the Nafion dryers effectively mitigate the impact of humidity variations on ozone measurements. Furthermore, while the relative humidity in the chamber did not frequently reach 100%, unlike ambient conditions, we did observe instances of high RH levels inside the chamber. However, it is essential to note that the potential impact of RH on our measurements has been effectively mitigated by the implementation of Nafion dryers. These dryers ensure the stability and accuracy of the measurements even under varying humidity conditions.**

figures 3 and 5: I suggest you indicate the rainy and sunny periods to help with the interpretation of the figure, and related text.

**The rainy periods have been highlighted in Figure 2. This positioning was chosen as it correlates with the weather data.**

figure 6: I suggest to increase the height of this figure and add the detection limit of TORM.

**DONE**

---

## Author Comment (AC3)

**Undetected BVOCs from Norway spruce drive total ozone reactivity measurements**

Steven Job Thomas[1,2], Toni Tykkä[1], Heidi Hellén[1], Federico Bianchi[2], and Arnaud P. Praplan[1]

[1]Atmospheric Composition Research, Finnish Meteorological Institute, Helsinki, 00101, Finland.
[2]Institute for Atmospheric and Earth System Research/Physics, Faculty of Science, University of Helsinki, Helsinki, 00560, Finland.

**Correspondence:** Steven Job Thomas (email: steven.thomas@helsinki.fi)

**Reviewer 1 comments and response**

In this revised manuscript, authors appropriately responded to reviewer's comments and the descriptions are improved and clarified. Especially, the new nomenclatures as 'TOZRE' and 'COZRE' are clear for readers to distinguish them. Therefore, this work is good enough to be published in this journal. Note that, for the revised points, the reviewer has to indicate a few technical suggestions.

1) Line 14 in Abstract:
To clarify and distinguish TOZRE and COZRE in Abstract, the following suggestion is possible.
(Now) However, the observed emissions …
(Suggestion) However, the observed emissions (COZRE) …

**Revised Version**

**However, COZRE made up only 35\% of the TOZRE, with sesquiterpenes being the most important sink for ozone.**

2) Lines 138-139:
Why does 'the continuous injection of ozone into the reactor counteract ozone decay' ? However, it is not clear.

Authors described in the answer to the previous comment, 'this underestimation arises due to the instrumental design …'. The reviewer guesses that such a counteraction is possible when the 'flasks' (not 'flow tubes') are utilized as the reactors. In the flasks, the decayed (decreased) ozone ($[O3](t)$) can be mixed with the continuous injection of ozone ($[O3](0)$) and the ozone decay can be 'diluted'. Is that right? If so, would you please clarify that such a counteraction was due to the reactor design (mixing in the flasks) as the following suggestion, for example?

(Now in Lines 138-139) The continuous injection of ozone into the reactor …
(Suggestion) In this study, flasks were utilized as the reactor. The continuous injection of ozone into the reactor (flasks) …
If the counteraction is due to other factors, would you please add a short description in the text?

**Revised Version:**

**TORM utilizes glass flasks as the reaction chamber unlike the flow tube reactor in Sommariva et al. (2020). The continuous injection of ozone into the reactor (flasks) counteracts…..**

**Undetected BVOCs from Norway spruce drive total ozone reactivity measurements**

Steven Job Thomas[1,2], Toni Tykkä[1], Heidi Hellén[1], Federico Bianchi[2], and Arnaud P. Praplan[1]

[1]Atmospheric Composition Research, Finnish Meteorological Institute, Helsinki, 00101, Finland.
[2]Institute for Atmospheric and Earth System Research/Physics, Faculty of Science, University of Helsinki, Helsinki, 00560, Finland.

Correspondence: Steven Job Thomas (email: steven.thomas@helsinki.fi)

**Reviewer 2 comments and response**

I think the authors have addressed most of the reviewers' concerns satisfactorily. I have two minor comments and, if they are satisfied, I think the paper can be accepted for publication.

1) With regard to Figure 8, and the associated discussion in section 3.5, it looks like a large fraction of the "high reactivity" data points actually overlap with the "low reactivity" data points. I would expect more, or better, sepration between the two groups. Can the authors comment on this?

**Certainly, your observation about the overlap in Figure 8 and the discussion in section 3.5 is valid. The overlap between "high reactivity" and "low reactivity" data points is due to the method of categorisation. We separated the periods based on daily averages: if the reactivity surpassed a specific threshold ($1 \times 10^{-9}$ $m^3$ $s^{-2}$ $g^{-1}$), it was marked as a high reactivity day; otherwise, it fell into the low reactivity category. Consequently, there are instances where reactivity and temperature coincide between days in both periods. For instance, reactivity might be similar during certain night-time and morning hours for both high and low reactivity periods. However, a clear separation exists between the two categories, supported by numerous data points indicating high reactivity.**

2) With regard to the calibration of TORM (appendix B1), I understand the authors' argument that the "two points" calibration has similar slope as the other calibration, but I still think it is not appropriate. I would suggest to consider the 5 points together in Figure B1 and show only one fitting line and equation. As I understand it (please confirm) the average slope is used throughout the paper anyways, so that would be more consistent in any case.

**Thank you for your comments. Indeed, the same coefficients were used throughout the paper. We have made corrections to the calibration graph and also the data. Since the values have changed a bit, all the calculations were redone.**

**Revisions:**
**Changes have been made to Figures 5, 6, 7, and 8 in the main content, along with Figure B1 in Appendix B1, incorporating updated values. Some values in the text have also been revised, although the differences were not significant. The data file has also been updated along with the link to the repository.**

---

## Author Response (AR3)

**Undetected BVOCs from Norway spruce drive total ozone reactivity measurements**

Steven Job Thomas[1,2], Toni Tykkä[1], Heidi Hellén[1], Federico Bianchi[2], and Arnaud P. Praplan[1]

[1]Atmospheric Composition Research, Finnish Meteorological Institute, Helsinki, 00101, Finland.
[2]Institute for Atmospheric and Earth System Research/Physics, Faculty of Science, University of Helsinki, Helsinki, 00560, Finland.

**Correspondence:** Steven Job Thomas (email: steven.thomas@helsinki.fi)

Images were modified to cater to individuals with colour blindness. Additionally, markers were strategically placed in specific areas to differentiate between variables.

The caption of figure 8 was modified slightly.

Previous version

TOZRE as a function of temperature for high reactivity period (red dots) and low reactivity period (green dots). Red and green lines indicate the exponential fit for the high and low reactivity periods respectively.

New version

TOZRE as a function of temperature for high reactivity period (orange dots) and low reactivity period (green dots). Red and green lines indicate the exponential fit for the high and low reactivity periods respectively.